# COVID-19 Vaccine Acceptance among an Online Sample of Sexual and Gender Minority Men and Transgender Women

**DOI:** 10.3390/vaccines9030204

**Published:** 2021-03-01

**Authors:** Daniel Teixeira da Silva, Katie Biello, Willey Y. Lin, Pablo K. Valente, Kenneth H. Mayer, Lisa Hightow-Weidman, José A. Bauermeister

**Affiliations:** 1Department of Medicine, School of Medicine, University of Pennsylvania, Philadelphia, PA 19104, USA; teixeird@pennmedicine.upenn.edu; 2Department of Family and Community Health, School of Nursing, University of Pennsylvania, Philadelphia, PA 19104, USA; willey@nursing.upenn.edu; 3Department of Behavioral and Social Sciences, School of Public Health, Brown University, Providence, RI 02912, USA; katie_biello@brown.edu (K.B.); pablo_valente@brown.edu (P.K.V.); 4Fenway Health, Boston, MA 02215, USA; kmayer@fenwayhealth.org; 5Beth Israel Deaconess Medical Center, Department of Medicine, Harvard Medical School, Boston, MA 02215, USA; 6Department of Medicine, School of Medicine, University of North Carolina—Chapel Hill, Chapel Hill, NC 27517, USA; lisa_hightow@med.unc.edu

**Keywords:** COVID-19, sexual, gender, minority, vaccine, acceptance

## Abstract

Sexual and gender minority (SGM) populations are particularly vulnerable to poor COVID-19 outcomes and are more likely to experience stigma and medical mistrust that may impact COVID-19 vaccine acceptance. This study examined the prevalence of COVID testing and diagnosis and assessed COVID-19 vaccine acceptance among a large sample of SGM. Participants were recruited as part of an online cross-sectional study focused on an HIV biomedical prevention technology willingness in the United States at increased risk for HIV sero-conversion. Multivariate linear analysis was conducted to examine COVID-19 vaccine acceptance. The study sample included 1350 predominately gay (61.6%), Black (57.9%), cis-gender (95.7%) males with a mean age of 32.9 years. Medical mistrust and social concern regarding COVID-19 vaccine stigma were significantly associated with decreased COVID-19 vaccine acceptance, and altruism was significantly associated with increased vaccine acceptance. Black participants were significantly less likely to accept a COVID-19 vaccine, and Asian participants were significantly more likely to accept a vaccine, compared to White peers. As the planning of COVID-19 vaccine rollout efforts is conceptualized and designed, these data may inform equitable implementation strategies and prevent worsening health inequities among SGM populations.

## 1. Introduction

Equitable implementation of COVID-19 vaccine delivery is a national and global priority. The Centers for Disease Control and Prevention Advisory Committee on Immunization Practice highlighted allocation strategies that “aim to reduce existing disparities and to not create new disparities” [1]. The National Academies of Sciences, Engineering, and Medicine published a framework for equitable allocation of COVID-19 vaccine that recognizes the rights and interests of sexual and gender minorities (SGM) but fails to identify strategies or data to achieve that goal [2]. Attitudes about COVID-19 vaccine acceptance can inform planning and implementation, and have been correlated with age, education, race, and employment status [3,4,5,6]. However, analyses have predominantly focused on cis-gender heteronormative populations, limiting their generalizability to SGM populations. Given that stigma and discrimination drive health inequities among SGM, which may result in increased risk of severe COVID-19 disease [7,8,9] and influence COVID-19 vaccine acceptance, research examining COVID-19 vaccine acceptance among SGM is needed. These data may inform equitable vaccine implementation strategies and prevent worsening health inequities among SGM populations.

The psychosocial and economic impact of the COVID-19 pandemic disproportionately affects SGM compared to cisgender heterosexual populations. Studies across international settings have demonstrated that SGM communities have experienced increased depression and anxiety as a result of social distancing measures and worrying about health status related to COVID-19 [10,11]. Studies examining global sample of SGM using a smart-phone based “Gay Social Networking” app found that since the beginning of the COVID-19 pandemic SGM have been more likely to experience job loss, income reduction, and decreased access to gender affirming resources [12,13]. In the United States, Latinx SGM have experienced increased personal violence due to stay-at-home orders, and racial/ethnic minority status has been associated with increased risk of severe COVID-19 [9,14,15]. Thus, COVID-19 related vulnerability likely varies across intersectional racial/ethnic, sexual, and gender minority identities. COVID-19 restrictions have been associated with decreased access to healthcare and decreased perceived social support, which may weaken resilience to minority stressors [13,16,17,18]. The confluence of disproportionate psychosocial and economic burdens with increased vulnerability to minority stressors among SGM may decrease COVID-19 vaccine uptake and further deepen health inequities.

Improved understanding of COVID-19 vaccine acceptance among SGM can inform equitable implementation of vaccine delivery strategies. In a study of HIV vaccine trial acceptance, for example, Connochie et al. demonstrated that greater vaccine efficacy beliefs and altruistic attitudes were associated with increased vaccine acceptance while social concerns (e.g., experiencing stigma and discrimination as a result of vaccination) decreased vaccine acceptance [19]. Studies examining human papillomavirus vaccine uptake among SGM have found increased acceptance associated with receiving provider recommendations and decreased acceptance associated with greater perceived barriers and lack of trust in providers [20,21]. Medical mistrust is engendered by systems that substantiate and reinforce racism, homophobia, and stigma and has been associated with decreased engagement in routine healthcare among SGM [22,23]. Trust in medical and scientific experts has been identified as a predictor of COVID-19 vaccine acceptance [24]. Perpetuating healthcare-related stigma in COVID-19 vaccine delivery strategies may lead to decreased vaccine acceptance and uptake among SGM, exacerbate health inequities, and threaten population level prevention of the COVID-19 vaccine. There is a lack of research, however, examining medical mistrust and stigma associated with COVID-19 acceptance among SGM. 

In a recent national survey representative of the United States population, only 53.6% of people planned to get the COVID-19 vaccine [25]. Understanding attitudes about COVID-19 vaccine acceptance is an urgent area of investigation to end the COVID-19 pandemic. The current study examined the prevalence of COVID testing and diagnosis and assessed COVID-19 vaccine acceptance among a large sample of SGM. We examined the associations between COVID-19 vaccine acceptance and medical mistrust, healthcare experiences, and attitudes towards COVID-19 vaccine. We hypothesized that increased levels of medical mistrust, negative healthcare experiences, and social concerns would be associated with decreased COVID-19 vaccine acceptance while altruism would be associated with increased acceptance among SGM in the United States.

## 2. Methods

### 2.1. Participants and Procedures

As part of an online cross-sectional study focused on a HIV biomedical prevention technology willingness in the United States, we included a series of COVID-19 vaccine acceptance and medical trust questions in our study screener. These questions were the focus of our analysis, and included data collected between October 19 and December 16, 2020 (e.g., prior to the FDA approval of the initial COVID-19 vaccines). Participants were primarily recruited through advertisements on social and sexual networking sites. There were no set regional or state geographic quotas for the sample.

### 2.2. Procedures

Upon clicking on the study advertisement, participants were referred to a Qualtrics survey where they were asked to consent and complete a study screener. Interested individuals completed a 10-min screener including questions regarding their sociodemographic characteristics, HIV and COVID testing behaviors, medical mistrust, and COVID-19 vaccine acceptance. Participants were not compensated for screening in the study. Study data were protected with a 256-bit SSL encryption and kept within a university firewalled server. The study was approved by the University of Pennsylvania Institutional Review Board, approval ID #843161.

### 2.3. Measures

COVID-19 Testing and Diagnosis. Participants were asked whether they had ever been tested for COVID-19 (0 = No; 1 = Yes) and whether they had been told by a health care provider that they had COVID-19 (0 = No; 1 = Yes).

COVID-19 Vaccine Acceptance. We assessed hypothetical efficacy based on scientific discussions prior to the results of the Pfizer-BioNTech and Moderna vaccines being released to the public and reviewed by the FDA. Specifically, participants were asked to rate how likely they would be to get a COVID-19 vaccine if it provided 85% protection. Participants used a 10-point scale (1 = “Extremely Unlikely”; 10 = “Extremely Likely”) to answer these questions. 

Attitudes towards COVID-19 vaccines. We assessed attitudes towards COVID-19 vaccines with two subscales previously developed by Lee et al. [26]. These subscales measure both altruistic attitudes towards getting vaccinated, as well as social concerns around getting vaccinated. The altruistic attitude subscale had four items addressing health promotive reasons in favor of becoming vaccinated (e.g., “My willingness to get a COVID-19 vaccine is important for the good of all people.”; α = 0.84), and the social concern scale had six items addressing many previously studied barriers to becoming vaccinated having to do with perceptions of others (e.g., “I would be concerned that getting a COVID-19 vaccine would lead to discrimination against me.”; α = 0.89). Both subscales asked participants to indicate their level of agreement with each statement presented, on a four-point scale (1 = Strongly Disagree; 2 = Disagree; 3 = Agree; 4 = Strongly Agree). We computed a mean score for each subscale.

Medical Mistrust. Using previously validated questions adapted from the Medical Mistrust Index [23,27]. We assessed participants’ agreement with three statements focused on participants’ medical mistrust (e.g., “Patients have sometimes been deceived or misled by health care providers”; “When health care providers make mistakes, they usually cover it up”; and “Health care providers have sometimes done harmful things to patients without their knowledge”). We also included three statements focused on medical trust (e.g., “I trust that health care providers are giving me the best treatment available”; “I trust that health care providers have my best interest in mind when treating me”; and “I trust that healthcare providers will tell me if a mistake is made about my medical treatment”). Participants were asked to indicate their level of agreement with each statement presented, on a four-point scale (1 = Strongly Agree; 2 = Agree; 3 = Disagree; 4 = Strongly Disagree). After reverse coding responses to statements focused on medical trust, we computed a mean composite score (α = 0.79) where higher scores meant greater medical mistrust.

In addition to mistrust beliefs, participants were asked to note how often they had felt that they had been mistreated or felt ignored by health care providers, or whether they had felt that their health care was not as good as others. Participants were asked to answer these three statements using a four-point scale (1 = Never; 2 = Rarely; 3 = Often; 4 = Always). We computed a mean composite score (α = 0.75) where higher scores meant greater frequency of negative experiences with health care providers in the past.

Sociodemographic characteristics. Participants were asked to report their age (in years), sex assigned at birth (Male, Female, or Other) and current gender identity (Male, Female, Transgender woman, Transgender Men, Gender variant/nonconforming, or Other). For analytical purposes, gender identity (0 = Cisgender Men; 1 = Gender Minority) was dichotomized due to small numbers within some of the gender identity categories in our regression analyses. Participants also self-identified across their races (Black, White, American Indian/Alaskan Native, Asian, Native Hawaiian or Other Pacific Islander, Multiracial, or Other) and Hispanic/Latinx ethnicity. Participants disclosed their sexual orientation (Gay/Homosexual, Bisexual, Heterosexual/Straight, Same Gender Loving, Queer, or Other).

## 3. Data Analytic Strategy

Descriptive statistics were conducted on demographics and variables of interest (Table 1). A multivariable linear regression framework was used to examine whether participants’ acceptance of a COVID-19 vaccine with 85% efficacy was associated with vaccine attitude subscales (e.g., altruism and social concern), medical mistrust, and prior negative experiences with health care providers (Table 2). We controlled for age, race/ethnicity, sexual orientation, gender identity, prior COVID-19 testing, and prior COVID-19 diagnosis in the regression analysis.

## 4. Results

As noted in Table 1, participants who consented to be screened and included in this analysis (N = 1350) had a mean age of 32.9 years (SD = 11.84). Our sample’s distribution by race was as follows: White (n = 349; 25.9%), Black (n = 752; 57.9%); Asian (n = 71; 5.3%), American Indian/Alaskan Native (n = 11; 0.8%), Native Hawaiian/Pacific Islander (n = 5; 0.4%), Multiracial (n = 77; 5.7%), or self-identified as Another Race (n = 55; 4.1%). Thirteen percent of the sample identified as Latinx (n = 180; 13.3%), with the majority identifying their ancestry as Puerto Rican (n = 64; 4.7%), Mexican (n = 45; 3.3%), Dominican (n = 31; 2.3%), or Cuban (n = 13; 1.0%). Most of the sample identified as cisgender men (n = 1292; 95.7%). The majority of gender minority participants were gender non-conforming (n = 28; 2.1%) or transgender female (n = 20; 1.5%). Participants identified predominantly as gay (n = 831, 61.6%) or bisexual (n = 229; 17.0%), while others identified as heterosexual (n = 85; 6.3%), queer (n = 25; 1.9%), same-gender loving (n = 24; 1.8%), another sexual identity (n = 18; 1.3%), or more than one sexual identity (n = 135; 10.0%). Nearly two-thirds of the sample reported ever having received a COVID-19 test (n = 821; 60.8%). Of those tested, 79 participants (9.6%) reported having been diagnosed with COVID-19 by a health care provider.

Acceptance for a COVID-19 vaccine providing 85% efficacy was moderately high among participants (mean score = 7, standard deviation = 3.12). In our multivariable regression (F (20, 1326) = 60.11, *p* < 0.001; see Table 2), participants’ acceptance of a COVID-19 vaccine was inversely associated with more social concerns regarding the vaccine (β = −0.10, *p* < 0.001) and medical mistrust (β = −0.06, *p* < 0.05). Vaccine acceptance was positively associated with altruistic motivations (β = 0.60, *p* < 0.001). In race comparisons, White participants were more willing to use a COVID-19 vaccine than Black, American Indian/Alaskan Native, and participants identifying with another race (see Table 2). Asian participants reported greater vaccine acceptance than White counterparts. Compared to gay men, participants grouped in the “Other” sexual identity category reported lower vaccine acceptance. There were no other differences by sexual identity. No association was found between acceptance of using an 85% efficacious COVID-19 vaccine and prior COVID-19 testing or diagnosis, age, gender identity, Latinx ethnicity, or negative experiences with providers in the past.

## 5. Discussion

Sexual and gender minority (SGM) populations are disproportionately vulnerable to poor COVID-19 outcomes, yet little is known about COVID-19 vaccine acceptance among SGM. To address this gap, we examined COVID-19 vaccine acceptance among SGM. As hypothesized, increased medical mistrust and social concerns were significantly associated with lower rates of vaccine acceptance, and altruism was associated with higher rates of vaccine acceptance. Our results highlight the impact of medical mistrust and attitudes toward COVID-19 vaccines on vaccine acceptance at the intersection of gender, sexual, and racial minority identities, and can inform equitable vaccine implementation strategies.

SGM who experienced medical mistrust were less likely to accept a COVID-19 vaccine. While negative experiences in healthcare were not associated with vaccine acceptance, medical mistrust may also stem from a long history of stigma and discrimination among SGM [8,23]. Medical mistrust is a driver of health inequity, and, in the context of the COVID-19 pandemic, may be promoted by conspiracy theories, disinformation, and misinformation [28]. Indeed, belief in COVID-19 conspiracy theories has been associated with medical mistrust as well as decreased likelihood of getting a COVID-19 vaccine [29]. Concerns about vaccine safety and side effects in the setting of expedited vaccine development and approval, and trust in government, may impact COVID-19 vaccine acceptance [3,30,31,32,33]. The recent promotion of misinformation by government leaders may have contributed to decreased vaccine acceptance and may have also reinforced medical mistrust of providers who refer to government guidelines and evidence to manage COVID-19 treatment and prevention [33]. In contrast to medical mistrust, COVID-19 vaccine acceptance was greater among study participants who endorsed altruistic attitudes and were less concerned about COVID-19 social concerns. Given that healthcare provider recommendations have been found to be associated with increased COVID-19 acceptance, these results suggest that providers and public health efforts that approach medical mistrust with empathy and validation address concerns about discrimination, and support altruistic intentions may be more successful engaging SGM in COVID-19 vaccine uptake [4,32]. While there is a lack of evidence-based interventions for improving trust in medicine, there are promising approaches available for future research [34]. For example, eHealth technology may be leveraged to increase trustworthiness and trust in medicine, and community-based participatory approaches can help design healthcare services that address medical mistrust [35,36]. In addition, lessons from AIDS denialism may have renewed relevance during the COVID-19 pandemic [28], and interventions effective in reducing HIV stigma, such as the Popular Opinion Leader model, may be adapted to address medical mistrust in SGM communities [37].

SGM participants in our study who identified as Black reported decreased COVID-19 vaccine acceptance. This finding is supported by prior studies [4,5,6,30]. Bogart and colleagues, for example, demonstrated medical mistrust to be associated with decreased COVID-19 vaccine acceptance among Black SGM [32]. Taken together, these findings underscore the need to address how systemic racism in the U.S. reduces economic opportunity, decreases healthcare access, and produces health inequities that are deepening in the setting of the COVID-19 pandemic [38]. For instance, racist rhetoric by government leaders in response to nationwide Black Lives Matter protests against police brutality may have contributed to decreased acceptance of COVID-19 vaccines promoted by federal agencies [32,39]. A national survey found that most Black adults agreed that government pandemic response would be stronger if more White people were affected [40]. Black adults also reported being less likely to get a COVID-19 vaccine even if it was free and determined safe by scientists, and cited concerns about safety and side effects more often than distrust in health systems [40]. Populations with intersecting minority racial, sexual, and gender identities may experience increased psychosocial stress compared to White SGM and Black heterosexual cis-gender populations [41]. Thus, our findings are likely indicative of psychosocial, economic, and structural factors impacting Black SGM and highlight how intersectionality impacts COVID-19 vaccine acceptance among SGM.

Our study has limitations that deserve to be mentioned. First, the study sample was an online-recruited national convenience sample of SGM interested in being screened for a HIV prevention study. While this approach allowed for an analysis of a large sample size, it may be preemptive to extrapolate the current findings to the larger population of SGM in the United States. Second, while the sample includes SGM from across the United States, there was no specific regional geographic quota set for the sample, limiting generalizability of our findings to the entire United States. Third, vaccine acceptance was assessed before results of the two FDA-approved COVID-19 vaccines were released to the public. As the public becomes more aware of the efficacy and safety of the COVID-19 vaccine, we may see shifts in SGM vaccine acceptance and intention to use it. Moreover, acceptance to adopt a COVID-19 vaccine may differ based on the vaccine regimen (e.g., dosage, time between treatment visits required), user characteristics and contexts, and treatment-related costs (e.g., out-of-pocket expenses, insurance). We encourage future research to apply sociobehavioral perspectives to examine how these vaccine-related considerations affect COVID-19 vaccine adoption. Fourth, our study sample included few gender minority individuals (n = 58), who may be particularly vulnerable to pandemic harms, and our results may not be generalizable to larger gender minority populations [42]. Lastly, our analysis compared outcomes between race categories among SGM populations, which limits our understanding of variability within racial groups that may help identify strategies to combat COVID-19 inequities. Moreover, in the absence of a racially-matched heterosexual sample, we are unable to assess how intersectional disparities across race, ethnicity, and sexual and gender identity could contribute to vaccine acceptance. Future research with larger population-based data may inform COVID-18 vaccine efforts.

Our analyses did not include measures of socioeconomic status, which can contribute to differences in COVID-19 vaccine acceptance, or other indicators of social vulnerability associated with increased risk of COVID-19 incidence and mortality [5,6,33,43]. Stigma, stress, and discrimination that contribute to health inequities among SGM likely impact vulnerability during the COVID-19 pandemic [7,8]. Structural factors such as high rates of unemployment and lack of health insurance among SGM that contributed to health inequity prior to COVID-19 are now worsening [12,13,44]. SGM are also more likely to experience homelessness that may increase exposure to COVID-19. Disproportionate rates of mental health disorders and victimization are worsening among SGM in the setting of social isolation due to pandemic restrictions [45,46,47,48]. Decreased access to health services and delays in seeking care may have grave consequences for SGM who are more likely to have comorbidities, such as asthma and cardiovascular disease, which increase risk of severe COVID-19 disease [9,17,44,48,49,50]. COVID-19 outcome data rarely include SGM identities, but co-occurring psychosocial, economic, and biomedical inequities among SGM increase risk for poor COVID-19 outcomes [7,9]. Public health surveillance must include “key equity indicators”, such as SGM identities, to ensure equitable representation in public policy [9,51]. Future research addressing medical mistrust may elucidate how experiences of social and economic inequality among SGM affect medical mistrust [28].

## 6. Conclusions

Our study suggests that medical mistrust, social concern, altruism, and race were significantly associated with COVID-19 vaccine acceptance among a national online sample of SGM. COVID-19 vaccine uptake is essential to stopping the spread of COVID-19 and ending the COVID-19 pandemic. As the planning and implementation of COVID-19 vaccine rollout efforts are designed, administrators must address challenges and opportunities related to SGM vaccine acceptance.

## Figures and Tables

**Table 1 vaccines-09-00204-t001:** Demographics of online sample of sexual and gender minority participants (N = 1350), October–December 2020.

	n (%) *
Mean Age (SD)	32.9 (11.84)
Race	
White	349 (25.9)
Black	752 (57.9)
Asian	71 (5.3)
American Indian/Alaskan Native	11 (0.8)
Native Hawaiian/Pacific Islander	5 (0.4)
Multiracial	77 (5.7)
Another race	55 (4.1)
Latinx ethnicity	180 (13.3)
Gender minority	58 (4.3)
Sexual orientation	
Gay	831 (61.6)
Bisexual	229 (17.0)
Queer	25 (1.9)
Same-gender loving	24 (1.8)
Multiple identities	135 (10.0)
Heterosexual	85 (6.3)
Another sexual identity	18 (1.3)

SD = standard deviation, * unless otherwise noted.

**Table 2 vaccines-09-00204-t002:** Multivariable linear regression model examining psychosocial correlates of COVID-19 vaccine acceptance among sexual gender minority participants, N = 1350.

	b	S.E.	B	t	*p*
(Constant)	1.035	0.523		1.978	0.048
Age	−0.003	0.005	−0.011	−0.549	0.583
Gender Minority	0.310	0.339	0.019	0.914	0.361
Latinx Ethnicity	0.021	0.198	0.002	0.107	0.915
Race/Ethnicity					
Black	−0.681	0.160	−0.108	−4.246	0.001
Asian	0.718	0.302	0.051	2.378	0.018
American Indian/Alaskan Native	−2.482	0.703	−0.072	−3.529	0.001
Native Hawaiian/Pacific Islander	−0.639	1.040	−0.012	−0.615	0.539
Multiracial	−0.112	0.295	−0.008	−0.382	0.703
Another Race	−1.174	0.348	−0.074	−3.374	0.001
Sexual Identity					
Bisexual	0.092	0.171	0.011	0.539	0.590
Queer	−0.162	0.469	−0.007	−0.345	0.730
Same-gender loving	−0.268	0.478	−0.011	−0.561	0.575
Multiple identities	−0.239	0.212	−0.023	−1.123	0.262
Other	−1.102	0.554	−0.041	−1.989	0.047
Social Concerns	−0.376	0.081	−0.100	−4.641	0.001
Altruistic Motivations	2.292	0.085	0.598	27.016	0.001
Medical Mistrust	−0.333	0.135	−0.060	−2.474	0.014
Negative Experiences with Provider	−0.073	0.112	−0.015	−0.652	0.515
Prior COVID-19 Test	0.024	0.131	0.004	0.186	0.852
Prior COVID-19 Diagnosis	0.203	0.271	0.015	0.749	0.454

Notes. Participants identifying as White/Caucasian serve as referent group for race comparisons. Participants identifying as gay serve as referent group for sexual identity.

## Data Availability

The data presented in this study are available on request from the corresponding author. The data are not publicly available due to privacy and ethical restrictions.

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
