# Peer review of "COVID-19 Vaccine Acceptance among an Online Sample of Sexual and Gender Minority Men and Transgender Women"

_vaccines, 2021, doi:10.3390/vaccines9030204_

Round 1
Reviewer 1 Report
This paper elaborates on the timely and important issue of SGM’s willingness to accept COVID-19 vaccine. Below are suggestions that might be helpful for revising the paper.
- Line 65: Since COVID-19 vaccine is not likely to cast any stigma to any of its receivers, compared with HIV vaccine, the stigma argument does not seem fit. It would be helpful if further supporting evidence or convincing argument is provided.
- Line 129: Are the measurement items for Medical Mistrust validated by prior research?
- Line 138: Because measurement items for both medical mistrust and trust contain positively worded statements, reverse coding all the positively worded statements could be misleading. For example, if a respondent strongly agrees with the item ‘Patients have sometimes been deceived or misled by health care providers’ and strongly disagrees with the item ‘I trust that health care providers are giving me the best treatment available’, which item gets reverse coded? Both items are positively worded.
- Line 132: harmful things
- Line 151: ‘For analytical purposes, gender identity (0=Cisgender Men; 1=Gender Minority) was dichotomized’ I’m not sure if I understand this correctly. Would then Cisgender Women be categorized as gender minority? A more detailed explanation would be helpful.
Author Response
Reviewer 1:
This paper elaborates on the timely and important issue of SGM’s willingness to accept COVID-19 vaccine. Below are suggestions that might be helpful for revising the paper.
- Line 65: Since COVID-19 vaccine is not likely to cast any stigma to any of its receivers, compared with HIV vaccine, the stigma argument does not seem fit. It would be helpful if further supporting evidence or convincing argument is provided.
Thank you for this comment. There is increasing evidence describing the impact of medical mistrust on health inequities, as well as understanding of how medical mistrust is formed. Jaiswal et al. published an article entitled “Towards a More Inclusive and Dynamic Understanding of Medical Mistrust Informed by Science” that defines medical mistrust as a phenomenon that is created in-part by stigma. Thus, medical mistrust experienced by SGM may be associated by experiences of stigma, which justifies our investigation of stigma related to the COVID-19 vaccine. The manuscript has been revised to include that argument in line 75, “Medical mistrust is engendered by systems that substantiate and reinforce racism, homophobia, and stigma” and cited Jaiswal et al.
- Line 129: Are the measurement items for Medical Mistrust validated by prior research?
The following sentence was added to the paragraph describing the study measure of medical mistrust, “Using previously validated questions adapted from the Medical Mistrust Index.” Citing two studies, LaVeist et al. and Eaton et al., that had previously validated the study measures.
- Line 138: Because measurement items for both medical mistrust and trust contain positively worded statements, reverse coding all the positively worded statements could be misleading. For example, if a respondent strongly agrees with the item ‘Patients have sometimes been deceived or misled by health care providers’ and strongly disagrees with the item ‘I trust that health care providers are giving me the best treatment available’, which item gets reverse coded? Both items are positively worded.
The paragraph describing the study measure of medical mistrust has been revised to be clearer. Line 143 specifies that participant responses to statements focused on medical trust were reverse coded.
- Line 132: harmful things
Thank you. Revised.
- Line 151: ‘For analytical purposes, gender identity (0=Cisgender Men; 1=Gender Minority) was dichotomized’ I’m not sure if I understand this correctly. Would then Cisgender Women be categorized as gender minority? A more detailed explanation would be helpful.
There were no cis-gender females in the study sample. Further description of participants categorized as gender minority was added to the results section, “The majority of gender minority participants were gender non-conforming (n=28; 2.1%) or transgender female (n=20; 1.5%).”

Reviewer 2 Report
Say "why" you need to investigate your population (e.g. stigma, mistrust, etc.) in your abstract. Stop repeating this in every section (e.g. your abstract should be different from your paper): "Equitable implementation of COVID-19 vaccine delivery is a national and global priority. Sexual and gender minority (SGM) populations are disproportionately vulnerable 208 to poor COVID-19 outcomes, yet little is known about COVID-19 vaccine acceptance 209 among SGM." Also, change paper to past tense. There are a lot of "is" in there. Remove "we" as well. Within the last section of the Discussion where you mention SES (which was what I was looking for in this paper!), please more deeply describe a component of future research in there. Finally, mention more specifically how vulnerable populations could be included and stigma and mistrust decreased re: the vaccine- aka, there is a lot of stigma research on different diseases, so maybe tie in an example from another global health problem. Overall, strong and interesting paper.
Author Response
Reviewer 2:
1) Say "why" you need to investigate your population (e.g. stigma, mistrust, etc.) in your abstract. Stop repeating this in every section (e.g. your abstract should be different from your paper): "Equitable implementation of COVID-19 vaccine delivery is a national and global priority. Sexual and gender minority (SGM) populations are disproportionately vulnerable 208 to poor COVID-19 outcomes, yet little is known about COVID-19 vaccine acceptance 209 among SGM."
The abstract has been revised to include that SGM populations are “are more likely to experience stigma and medical mistrust that may impact COVID-19 vaccine acceptance.” The manuscript was revised to not repeat the above phrase.
2) Also, change paper to past tense. There are a lot of "is" in there. Remove "we" as well.
Revised manuscript throughout.
3) Within the last section of the Discussion where you mention SES (which was what I was looking for in this paper!), please more deeply describe a component of future research in there.
The last paragraph of the manuscript has been revised to include the following sentence, “Public health surveillance must include “key equity indicators”, such as SGM identities, to ensure equitable representation in public policy. Future research addressing medical mistrust may elucidate how experiences of social and economic inequality among SGM affect medical mistrust.”
4) Finally, mention more specifically how vulnerable populations could be included and stigma and mistrust decreased re: the vaccine- aka, there is a lot of stigma research on different diseases, so maybe tie in an example from another global health problem.
Thank you for this comment. The following text has been added to the discussion, “While there is a lack of evidence-based interventions for improving trust in medicine, there are promising approaches for future research. For example, eHealth technology may be leveraged to increase trustworthiness and trust in medicine and community-based participatory approaches can help design healthcare services that address medical mistrust. In addition, lessons from AIDS denialism may have renewed relevance during the COVID-19 pandemic, and interventions effective in reducing HIV stigma, such as the Popular Opinion Leader model, may be adapted to address medical mis-trust in SGM communities.”
Overall, strong and interesting paper.

Reviewer 3 Report
The work presented in this manuscript is on the acceptance of COVID-19 vaccine among SGM population. The topic is up-to-date and interesting; however, the purpose and design of the study is somewhat unclear.
The authors’ hypothesis (line 86-89) is quite reasonable regardless the study subjects are SGM or non-SGM people. However, it is not clear at all why do the authors need to investigate this only among SGM people. It is not clear how do the authors expect this reasonable hypothesis affected/different among SGM people. I do sympathize that SGM people may confront disproportion in socioeconomic, medical, or etc. impact of COVID-19 pandemic; however, how the authors hypothesize that the situation affects SGM people’s attitude to vaccine?
The authors are required to make this point clear in the revised manuscript.
It would have been more fruitful if the present results could be compared with the results of non-SGM people. This would have extracted the difference in attitude to vaccine between SGM and non-SGM people, if present, that could have been helped policy making for SGM people. This is a problem of study design and the authors cannot correct at this time.
Line 52-54. More detailed descriptions may be necessary about refs. 10-14 as to how these psychosocial and socioeconomic aspects disproportionally affected SGM people for the international readers of this journal. Generally, the authors are encouraged to explain more about the situations in the USA in this manuscript because there are many readers of other countries who are not familiar with USA situations.
Line 106. Specify the affiliation of IRB and approval ID number for this study.
Author Response
Reviewer 3:
The work presented in this manuscript is on the acceptance of COVID-19 vaccine among SGM population. The topic is up-to-date and interesting; however, the purpose and design of the study is somewhat unclear.
1) The authors’ hypothesis (line 86-89) is quite reasonable regardless the study subjects are SGM or non-SGM people. However, it is not clear at all why do the authors need to investigate this only among SGM people. It is not clear how do the authors expect this reasonable hypothesis affected/different among SGM people. I do sympathize that SGM people may confront disproportion in socioeconomic, medical, or etc. impact of COVID-19 pandemic; however, how the authors hypothesize that the situation affects SGM people’s attitude to vaccine?
The authors are required to make this point clear in the revised manuscript.
Thank you for this comment. In addition to socioeconomic and structural factors, medical mistrust among SGM is likely to be formed in the context of experiencing stigma and homophobia. The manuscript has been reviewed to clearly describe the relationship between medical mistrust, stigma, homophobia and engagement in healthcare among SGM. The following sentence is now included in line 74-77 “Medical mistrust is engendered by systems that substantiate and reinforce racism, homophobia, and stigma, and has been associated with decreased engagement in routine healthcare among SGM.” This argument highlights that medical mistrust is created in an environment of stigma and homophobia that is more likely to impact SGM populations further justifying the investigation of the association between medical mistrust and COVID-19 vaccine acceptance among SGM.
2) It would have been more fruitful if the present results could be compared with the results of non-SGM people. This would have extracted the difference in attitude to vaccine between SGM and non-SGM people, if present, that could have been helped policy making for SGM people. This is a problem of study design and the authors cannot correct at this time.
The following sentence was added to the limitations paragraph, “Moreover, in the absence of a racially-matched heterosexual sample, we are unable to assess how intersectional disparities across race, ethnicity, and sexual and gender identity could contribute to vaccine acceptance.”
3) Line 52-54. More detailed descriptions may be necessary about refs. 10-14 as to how these psychosocial and socioeconomic aspects disproportionally affected SGM people for the international readers of this journal. Generally, the authors are encouraged to explain more about the situations in the USA in this manuscript because there are many readers of other countries who are not familiar with USA situations.
The references cited above are not limited to the USA. As described in the manuscript by the phrase “Studies across international settings,” these references include study populations from Brazil, Hong Kong, and global samples. The following text was added to describe the studies that include a global sample, “Studies examining a global sample of SGM using a smart-phone based “Gay Social Networking” app.”
4) Line 106. Specify the affiliation of IRB and approval ID number for this study.
The following statement was added to the procedures paragraph, “The study was approved by the University of Pennsylvania Institutional Review Board, approval ID #843161.”

Round 2
Reviewer 3 Report
The authors have revised the manuscript according to the reviewer's comments.